# Effects of Korean red ginseng on three-dimensional trabecular bone microarchitecture and strength in growing rats: Comparison with changes due to jump exercise

**Yong-In Ju[1]\*, Hak-Jin Choi[2], Teruki Sone[3]**

**1** Department of Health and Sports Sciences, Kawasaki University of Medical Welfare, Kurashiki, Okayama, Japan, **2** School of Sport for All, Kyungwoon University, Gumi, Republic of Korea, **3** Department of Nuclear Medicine, Kawasaki Medical School, Kurashiki, Okayama, Japan

\* jyi.k051@mw.kawasaki-m.ac.jp

**Data Availability Statement:** All relevant data are within the paper and its Supporting information files.

## Abstract

### Objectives

The preventive effects of Korean red ginseng (KRG) on bone loss and microarchitectural deterioration have been extensively studied in animal models. However, few results have been reported for the effects of KRG on the trabecular microarchitecture as compared to changes resulting from physiological stimuli such as exercise load. We compared the effects of KRG and jump exercise on improvements in trabecular microarchitecture and strength of the distal femoral metaphysis in rats.

### Methods and materials

Eleven-week-old male Wistar rats were divided into sedentary (CON), KRG-administered (KRG), and jump-exercised (JUM) groups. Rats were orally administered KRG extract (200 mg/kg body weight/day) once a day for 6 weeks. The jump exercise protocol comprised 10 jumps/day, 5 days/week at a jump height of 40 cm. We used microcomputed tomography to assess the microarchitecture, volumetric bone mineral density (vBMD), and fracture load as predicted by finite element analysis at the right distal femoral metaphysis. The left femur was used for the quantitative bone histomorphometry measurements.

### Results

Although KRG produced significantly higher trabecular bone volume (BV/TV) than CON, BV/TV was even higher in JUM than in KRG, and differences in vBMD and fracture load were only significant between JUM and CON. In terms of trabecular microarchitecture, KRG increased trabecular number and connectivity, whereas the JUM group showed increased trabecular thickness. Bone resorption showed significant decrease by JUM and KRG group. In contrast, bone formation showed significant increase by JUM group.

**Funding:** This study was supported by Grant-in-Aid for Scientific Research (C) No. 15K01738 from the Japan Society for the Promotion of Science.

**Competing interests:** The authors have declared that no competing interests exist.

## Conclusions

These data show that KRG has weak but significant positive effects on bone mass and suggest that the effects on trabecular microarchitecture differ from those of jump exercise. The effects of combined KRG and jump exercise on trabecular bone mass and strength should be investigated.

## Introduction

The achievement of as maximal a bone mass and bone strength as possible during growth is considered to be the best protection against age-related bone loss or osteoporosis and the subsequent risk of osteoporotic fractures [1]. Physical activity, nutrition diet, and other lifestyle factors can greatly impact bone health in an adolescent. Regular exercise and a balanced diet during childhood and adolescence, in general, is known to be the major factor affecting maximum achievement of bone mass. Thus, exercise and healthy nutrition during the early stage of life may be an effective countermeasure to prevent bone fracture by osteoporosis and age-related osteopenia.

The ginseng plant (*Panax ginseng Meyer*) has been considered a medicinal plant in Asian countries including Korea, China and Japan for thousands of years. This plant is now in wide use as an alternative medicine for the treatment of disease and the promotion of health. Numerous research studies have confirmed the biological activities of ginseng, such as anti-inflammatory [2], anti-oxidative [3, 4], anti-obesity [5], anti-diabetic [6], anti-stress [7], and anti-cancer properties [8, 9]. In general, the commercially available ginseng in South Korea can be broadly categorized as fresh ginseng, white ginseng, or Korean red ginseng (KRG). KRG is made using a process of repeatedly steaming and air-drying fresh raw ginseng without peeling, while white ginseng is produced by sun-drying. KRG is known to have more pharmacological effects and stability compared with fresh and white ginseng, because of the changes to the chemical components (e.g., ginsenosides Rg2, Rg3, Rh1, and Rh2) produced in the steaming process [10]. KRG is an important component in medicine and various health supplements, providing antiaging, antioxidant, and immunomodulatory effects with a low rate of side effects [11, 12]. However, compared with the widespread research on the biological activity of KRG, studies on the responses of bone to KRG remain limited. Several recent studies have shown that KRG and ginsenoside has the ability to inhibit bone loss in the ovariectomy-induced osteoporotic rodent model [13, 14], and to inhibit bone resorption in cell cultures [15–18]. However, little information has been accumulated regarding the effects of KRG on bone microarchitecture and strength. In addition to low bone mass, deterioration of bone microarchitecture is an important determinant of bone fragility in osteoporosis. Evaluation of changes in bone microarchitecture by KRG and determination of its significance in terms of improving bone strength are therefore important.

Physical exercise has been considered suitable methods to help increase in bone mass during growth and to prevent age-related bone loss. However, not all types of exercise exert the same beneficial effects on the skeleton. Among various types of exercise in rat models, jump exercise is one of the most effective methods for increasing bone mass and strength [19–21]. In a previous study, we confirmed that jump exercise can have positive effects on trabecular bone strength via changes to the trabecular thickness (microarchitecture) of the distal femoral metaphysis in growing rats [22]. In contrast, previous research has suggested that administration of KRG extract to irradiated mice prevented loss of trabecular bone mass of the proximal

tibia, primarily by significant alterations in trabecular number [23]. KRG administration and jump exercise may thus show different effects on trabecular microarchitecture. To the best of our knowledge, no previous studies have directly compared such effects between KRG administration and exercise, to assess the types and degrees of difference. The objective of this study was thus to compare the effects of exercise and KRG on trabecular bone architecture and to assess the influence of KRG as an effective strategy for improving bone strength during growth. The reason for selecting jump exercise for comparison was that structural changes to bone caused by the physiological stimulation of jump exercise are considered representative of ideal change [19–21].

## Materials and methods

### Ethics statement and euthanasia

This study was approved by the Animal Ethics Committee of the Kawasaki University of Medical Welfare (Permit Number: 15–005). The experiment protocol and all animal care described in this investigation were performed in accordance with the Guidelines for Animal Experiments of the Kawasaki University of Medical Welfare. Each rat was euthanized using a combination of medetomidine hydrochloride (0.15 mg/kg, Dorbene® Vet; Kyoritsu Seiyaku Corporation, Tokyo, Japan), midazolam (2 mg/kg, Sandoz, Yamagata, Japan), and butorphanol (2.5 mg/kg, Vetorphale; Meiji Seika Pharma Co., Tokyo, Japan) administered via intraperitoneal injections at the end of the experiment, and all possible efforts were made to minimize the rat's suffering.

### Animal and experimental design

Thirty male Wistar rats were used in this study. Animals were obtained from Clea Japan (Osaka, Japan) at 10 weeks old. All rats were singly housed in standard cages (20 × 33 × 14 cm) under a constant temperature of 22 ± 1˚C and a 12:12 light-dark cycle. Rats were given ad libitum access to standard laboratory animal chow (MF; Oriental Yeast, Chiba, Japan) containing 1.15% calcium and 0.88% phosphorus and water. Rats were habituated to the diet and new environment for 1 week. The health and behavior of animals were monitored at least twice daily. After 7 days of acclimation, rats were randomly assigned into three groups (n = 10 each) as follows: a sedentary control group (CON); a group administered KRG (KRG); and a jump exercise group (JUM). All rats were double-labeled with subcutaneous injections of 10 mg per kg of fluorescent calcein (Dojindo, Kumamoto, Japan) at 1 and 5 days before sacrifice. Soon after euthanasia, right calf muscles were collected from each rat and immediately weighed. Excised femora from each rat were cleaned of soft tissue, then femoral length was measured using digital calipers. Right femora were stored at -40˚C until analysis using micro-computed tomography (micro-CT) and left femora were fixed in 70% ethanol for histomorphometric analysis.

### Exercise protocol

We applied the jump exercise protocol we have used previously and the details have been described elsewhere [22, 24–26]. Briefly, rats in the JUM group were individually placed at the bottom of a wooden box, 40 cm × 40 cm and 40 cm high. The initial height of the box was 10 cm, and this was gradually increased to 40 cm during the first week. Since rat grasped the top of the board set at 40 cm with their forelimbs and then climbed onto it, the true jump height is estimated to be 30cm to 35cm. An electric stimulus was initially provided to force them to jump and to grasp the top of one of the sides of box with the forelimbs. The rat was then

returned to the floor of the box for the next jump. The jump exercise program comprised 10 times per day, 5 days per week for 6 weeks. We improved the training environment as favorable as possible so that the rat can jump without much stress. For example, each jump exercise session was performed during the dark period at the same time each day. More importantly, since the rats have been conditioned to jump voluntarily soon after the start of exercise training, the electrical stimulus was required only in the initial few days. Even when the rat has interrupted the jump, light tap on the cage wall by a technician can restart the continuous jumping.

## KRG preparation and administration

The KRG extract in concentrated form used in this study was purchased commercially from the Korea Ginseng Corporation ("Cheong Kwan Jang", root extract of 6-year-old fresh KRG; Daejeon, Republic of Korea). The KRG extract dose was selected on the basis of previously published data [27]. The KRG extract was freshly prepared by dissolving in distilled water before administration. Each rat in the KRG group was orally administered KRG 5 days/week for 6 weeks at a dose of 200 mg/kg body weight/day of KRG solution dissolved in distilled water. KRG extract contained major ginsenoside-Rb1: 5.61 mg/g, -Rb2: 2.03 mg/g, -Rc: 2.20 mg/g, -Rd: 0.39 mg/g, -Re: 1.88 mg/g, -Rf: 0.89 mg/g, -Rg1: 3.06 mg/g, -Rg2s: 0.15 mg/g, -Rg3s: 0.17 mg/g, -Rg3r: 0.08 mg/g, and other minor ginsenosides. Rats in CON and JUM groups were orally administered volumes of distilled water equal to the volume of KRG solution. Oral administration was performed using an oral sonde to ensure the correct doses.

## Micro-CT scanning

The bone microarchitecture and volumetric bone mineral density (vBMD, mg/cm$^3$) of the right femur were analyzed by Ele Scan mini micro-CT system (Nittetsu Elex, Tokyo, Japan) as previously described [22, 24–26]. Measurement conditions for micro-CT were as follows: source energy, 30 kVp; electric current, 80 μA; filter, 0.1-mm copper plate; slice thickness, 18.11 μm; matrices, 512 × 512; pixel size, 18.11 μm. The sample area at the distal femoral metaphysis were scanned in an area 2.8–3.0 mm proximal to the distal end of the femur, including the border between the distal metaphysis and growth plate (Fig 1B). A total of 300 consecutive slices (5.4 mm) were taken for analysis of trabecular bone, with the region of interest was defined as the 130 slices (2.4 mm) above the most proximal portion of the growth plate (Fig 1C). After micro-CT scanning, the original image data were transferred to a workstation and analysed using 3D image analysis software (TRI/3D-BON; Ratoc System Engineering, Tokyo, Japan). Trabecular and cortical bone were separated (Fig 1D), and the following structural indices were calculated: trabecular bone volume fraction (BV/TV, %), trabecular thickness (Tb.Th, μm), trabecular number (Tb.N, 1/mm), trabecular separation (Tb.Sp, μm), connectivity density (Conn.D, 1/mm$^3$), trabecular bone pattern factor (TBPf, 1/mm), and structure model index (SMI) [28]. In addition, vBMD was determined using TRI/3D-Bon BMD software (Ratoc System Engineering, Tokyo, Japan). The densitometer's BMD values were calibrated with a hydroxyapatite phantom (6 × 1 mm; 200, 300, 400, 500, 600, 700 and 800 mg/cm$^3$; Kyoto Kagaku, Kyoto, Japan). The calibration phantom was scanned in the same conditions as actual bone.

## Finite element analysis (FEA) for micro-CT images

Micro-FEA were performed using TRI/3D-FEM64 software (RATOC System Engineering, Tokyo, Japan). Reconstructed 3D images of the distal femur obtained by micro-CT (130 slices with a voxel size of 18.11 μm) were exploited for FEA. Compression loading condition was

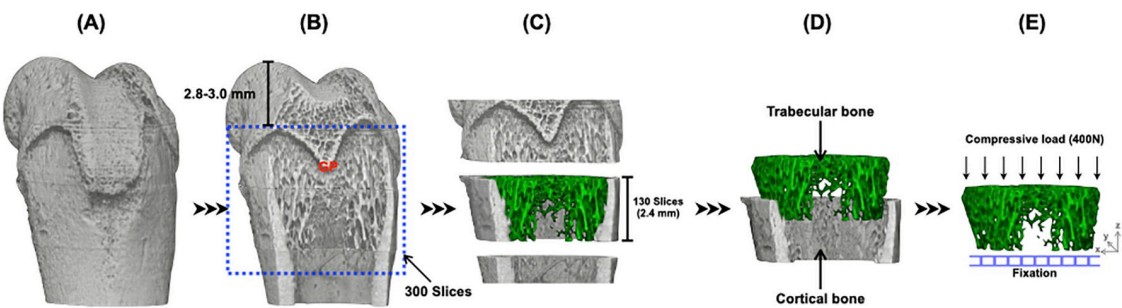

**Fig 1. Micro-CT analyses of trabecular bone microarchitecture in the distal femoral metaphysis.** A) Representative 3D images of the femoral trabecular bone architecture scanned by micro-CT. B) The distal femoral metaphysis was scanned in an area 2.8–3.0 mm proximal to the distal end of the femur, including the border between the metaphysis and growth plate (GP). C) The region of interest selected for analysis, including cortical and trabecular bone of the distal femoral metaphysis (130 slices). D) Cortical and trabecular bone were subsequently separated and 3D structural indices were calculated. E) After analyzing the trabecular bone architecture, samples were subjected to FEA for prediction of fracture loads. The nodes located at the proximal surface of the femur was fully fixed in all directions, and compression test was simulated at a load of 400N applied to the distal surface of the femur.

simulated by fully constraining the nodes located at the proximal surface of the femur and a distributed load was applied on the distal surface along the perpendicular direction (Fig 1E). The magnitude of the applied force was chosen to be 400 N. The trabecular bone tissue was modeled as an isotropic, linear elastic material. Young's modulus of the trabecular bone was calculated based on Carter's equation and mineralization value from micro-CT images [29] according to the following equation: $E = 16.311 (-5) \times$ [trabecular mineralization density (mg/ cm$^3$)]$^3$. Where E is the Young's modulus (measured in MPa). Poisson's ratio was set to the regular value of 0.3. The element on this surface was restrained to one voxel. Fracture load was defined as the load when 2.8% of the elements reached a shear stress of 68 Mpa or more. Concurrently, the total reaction force on the bottom surface (= fixed area) was analyzed as a fracture load.

## Bone histomorphometry

After sacrifice, the left femur was isolated, fixed in 70% pure ethanol, and embedded in methyl-methacrylate (Wako Pure Chemical Industries, Osaka, Japan) without bone decalcification. Frontal plane sections (5 μm thick) of the distal metaphysis were cut using a RM2255 rotary microtome (Leica, Wetzlar, Germany) and stained with Villanueva osteochrome bone stain (basic fuchsin, fast green, orange G, and azure II; Merck, Darmstadt, Germany) for bone histomorphometry as previously reported [24]. Static and dynamic histomorphometric analyses were measured at the Ito Bone Histomorphometry Institute (Niigata, Japan). The following parameters were measured using a semiautomatic graphic system (Histometry RT CAMERA; System Supply Co., Nagano, Japan): BV/TV (%); Tb.Th (μm); Tb.N (N/mm); Tb.Sp (μm); osteoclast surface per bone surface (Oc.S/BS, %); eroded surface per BS (ES/BS, %); number of osteoclasts per BS (N.Oc/BS, N/mm); number osteoblast surface per BS (Ob.S/BS, %); mineralizing surface per BS (MS/BS, %); number of osteoblasts per BS (N.Ob/BS, N/mm); bone formation rate per BS (BFR/BS, μm$^3$/μm$^2$/day), and mineral apposition rate (MAR, μm/day) [30].

## Data analysis

Data analysis was performed using IBM SPSS Statistics version 26.0 software package (SPSS, Chicago, IL). The experimental results are expressed in the form of the mean and standard deviation (SD). Data were tested for normal distribution using the Shapiro-Wilk test and

**Table 1. Body weight, hindlimb muscle weight, and femoral length in experimental rats.**

|  | CON | KRG | JUM |
|---|---|---|---|
| Body weight before experiment (g) | 309.10 ± 17.15 | 304.60 ± 08.18 | 301.30 ± 09.82 |
| Body weight after experiment (g) | 423.80 ± 31.16 | 419.20 ± 20.78 | 422.00 ± 24.19 |
| Calf muscle weight (g) | 2.03 ± 0.16 | 2.03 ± 0.12 | 2.05 ± 0.11 |
| Femoral length (mm) | 36.31 ± 0.57 | 36.41 ± 0.37 | 35.73 ± 0.70 |

Values are shown as mean ± SD. Male Wistar rats that were 11 weeks old at the beginning of the study were distributed into 3 groups: a group administered 200 mg/kg body weight of Korean red ginseng daily (KRG); a group that jumped 10 times/day from a height of 40 cm (JUM); and control group without exercise or KRG administration (CON).

homogeneity of variance using Levene's test. If data failed these tests, the significance of differences across groups was evaluated using the Kruskal-Wallis nonparametric test. For the variables that exhibited normal distribution, one-way ANOVA and Tukey's post hoc tests were used for the statistical analysis of significance. Values of $p < 0.05$ were considered statistically significant.

# Results

## Physical parameters of rats

Body weight before and after the experiment, calf muscle weight and femoral length of rats from each group are shown in Table 1. During the course of the experiment, no differences in body weight were noted among the CON group, JUM group, and KRG group. Calf muscle weight and femoral length did not significantly differ among the CON, JUM, and KRG groups at the end of the experiment.

## Microstructural properties, vBMD and FEA-predicted fracture load

Results for 3D microstructural parameters and vBMD in the distal metaphysis of the femur are shown in Fig 2. In terms of femoral trabecular bone parameters, BV/TV, Tb.Th, and Tb.N were significantly higher (72%, $p<0.001$; 51%, $p<0.001$; and 14%, $p<0.001$, respectively) and Tb.Sp, SMI, TBPf, and DA were significantly lower (-23%, $p<0.001$; -66%, $p<0.001$; -75%, $p<0.001$; and -6%, $p<0.001$, respectively) in the JUM group compared with the CON group. In the KRG group, BV/TV, Tb.N, and Conn.D were significantly higher than in the CON group (17%, $p<0.05$; 13%, $p<0.001$; and 17%, $p<0.05$, respectively). In contrast, BV/TV and Tb.Th were significantly higher (48%, $p<0.001$; and 46%, $p<0.001$, respectively), whereas DA was significantly lower in the JUM group than in the KRG group (-5%, $p<0.01$). A significant increase in vBMD was seen in the JUM group compared to both CON and KRG groups (118%, $p<0.001$ and 55%, $p<0.001$, respectively). Finally, FEA-predicted fracture loads were significantly higher in the JUM group than in the CON and KRG groups (196%, $p<0.01$ and 108%, $p<0.05$, respectively) (Fig 3).

Figs 4 and 5 show representative 3D trabecular microarchitecture and histological images in the distal femoral metaphysis for a rat from each group. Both jump-exercised and KRG-administered rats demonstrated that diffuse increases in trabecular bone and differential effects of jump exercise as compared to KRG on trabecular microarchitecture can be confirmed visually from these images.

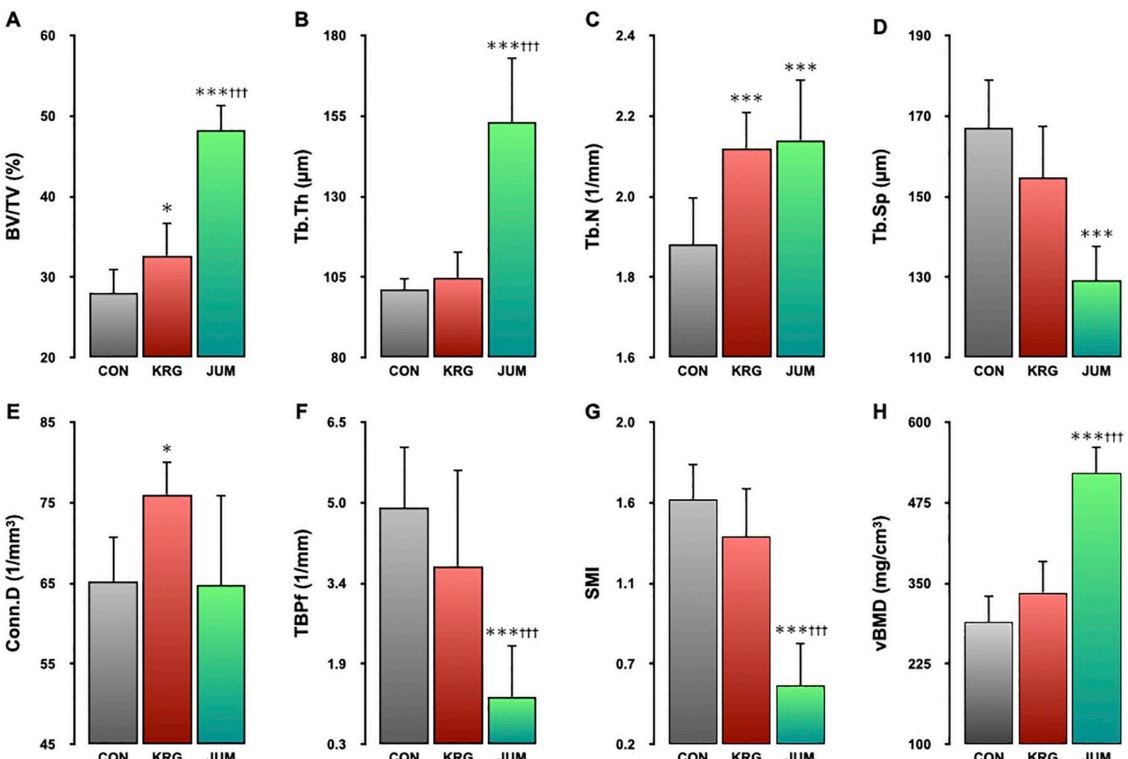

**Fig 2. Microstructural parameters in the distal femoral metaphysis as measured by micro-CT.** Trabecular microarchitecture parameters of the distal femoral metaphysis are shown as: BV/TV, trabecular bone volume fraction (A); Tb.Th, trabecular thickness (B); Tb.N, trabecular number (C); Tb.Sp, trabecular separation (D); Conn.D, connectivity density (E); TBPf, trabecular bone pattern factor (F); SMI, structure model index (G); vBMD, volumetric bone mineral density (H). Male Wistar rats that were 11 weeks old at the beginning of the study were distributed into 3 groups: a group administered 200 mg/kg body weight of Korean red ginseng daily (KRG); a group that jumped 10 times/day from a height of 40 cm (JUM); and control group without exercise or KRG administration (CON). All data represent mean ± SD. Significant difference vs. CON group: $^*p<0.05$; $^{**}p<0.01$; $^{***}p<0.001$. Significant difference vs. KRG group: $^{††}p<0.01$; $^{†††}p<0.001$.

## Histomorphometry

Results for static and dynamic histomorphometric at the distal femoral metaphysis of rats from each group are shown in Table 2. Regarding static histomorphometry, BV/TV and Tb.N in both JUM (85%, $p<0.001$; and 57%, $p<0.001$, respectively) and KRG groups (57%, $p<0.01$; and 37%, $p<0.01$, respectively) were significantly higher than in the CON group, whereas Tb.Sp was significantly lower in the JUM (-44%, $p<0.001$) and KRG groups (-32%, $p<0.001$) than in the CON group. Conversely, Tb.Th was highest in the JUM group and differed significantly from that in the CON group. With respect to bone resorption, Oc.S/BS, ES/BS, and N.Oc/BS in both JUM (-30%, $p<0.05$; -33%, $p<0.01$; and -31%, $p<0.05$, respectively) and KRG groups (-36%, $p<0.01$; -20%, $p<0.05$; and -42%, $p<0.001$, respectively) were significantly lower than in the CON group. In contrast, values for N.Ob/BS reflecting bone formation were higher in the JUM group than in the CON and KRG groups (p = 0.363 and p = 0.688, respectively).

## Discussion

Our primary focus in this study was to compare the effects of KRG administration on trabecular microarchitecture with changes caused by jump exercise. We found that jump exercise and KRG administration exerted different effects on trabecular microarchitecture in the distal

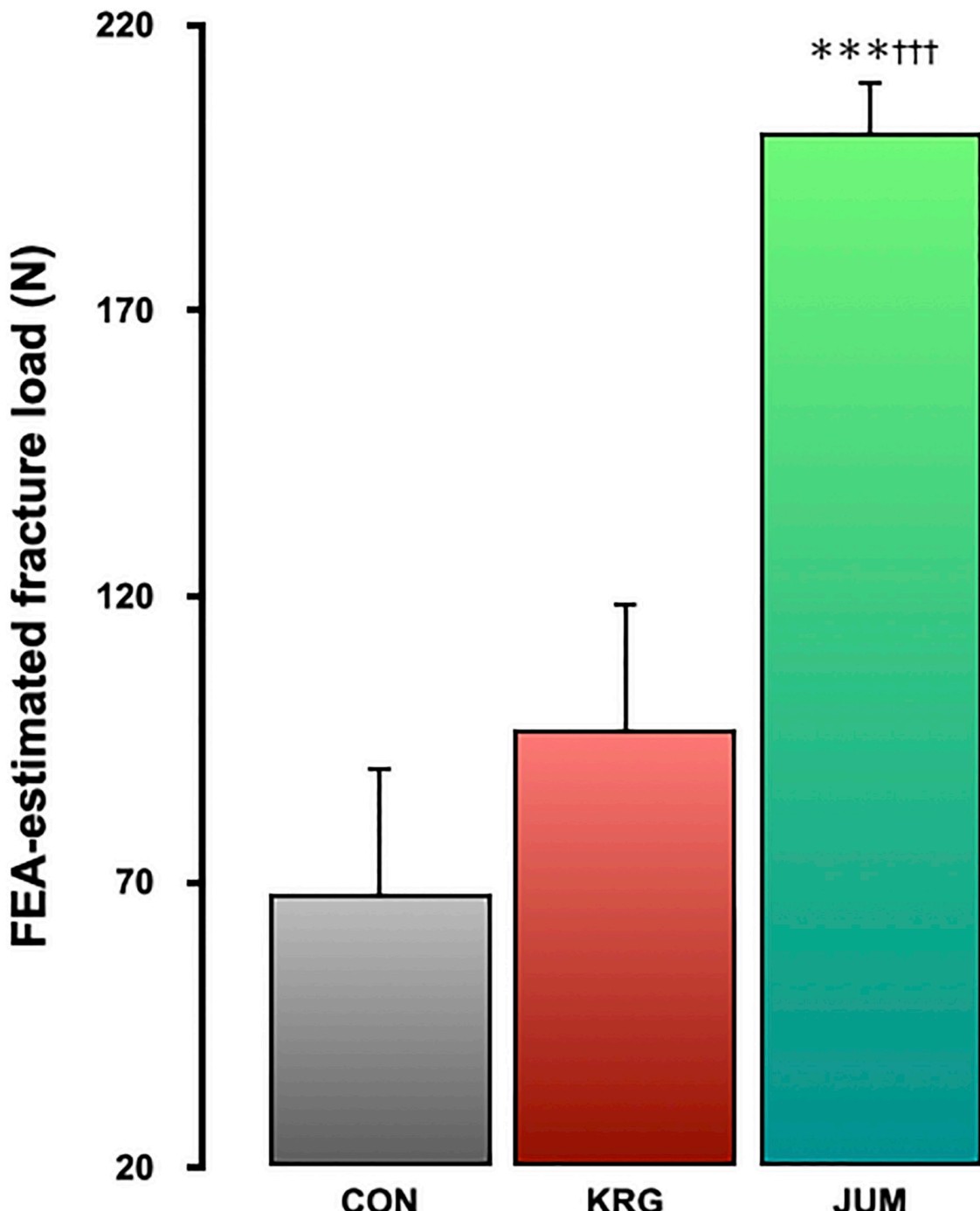

**Fig 3. Fracture load as predicted by FEA.** CON, sedentary control group; KRG, KRG administration group; JUM, jump exercise group. All data represent mean ± SD. Significant difference vs. CON group: ***$p < 0.001$. Significant difference vs. KRG group: †††$p < 0.001$.

femoral metaphysis of growing male rats. Namely, jump exercise improved both trabecular thickness and number, whereas KRG predominantly improved trabecular number.

The effects of exercise on bone mass and strength depend on loading magnitude and strain rate applied to the bone during the activity. A higher strain rate is associated with a greater

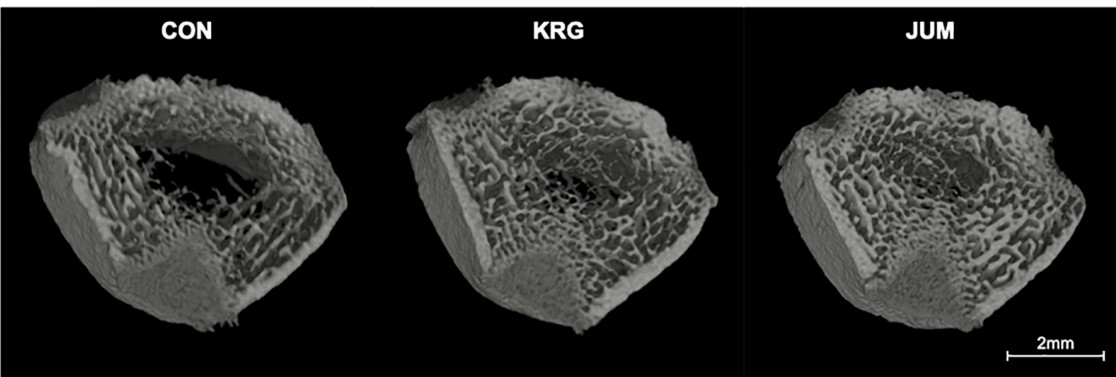

**Fig 4. Representative three-dimensional micro-CT images of trabecular bone of the distal femoral metaphyseal region from each group were reconstructed.** KRG, group administered 200 mg/kg body weight of Korean red ginseng daily; JUM, group that jumped 10 times/day from a height of 40 cm; CON, control group without exercise or KRG administration.

adaptive bone response [31]. Several studies with rats have found that jumping yields greater increases in bone mass and strength because of the greater mechanical stress and higher strain rate imposed [19–21]. Previous histomorphometric analyses have shown that the increase in trabecular bone mass that occurs with resistance exercise is primarily due to increased trabecular thickness, rather than noticeable changes in numbers of trabeculae [32, 33]. In previous experiments with 3D micro-CT of the distal femoral metaphysis, we observed that the gain in trabecular bone induced by jump exercise during the remobilization period was also predominantly attributable to increases in trabecular thickness rather than number [25]. The present study showed similar results, with a 72% increase in trabecular bone mass, a 51% increase in

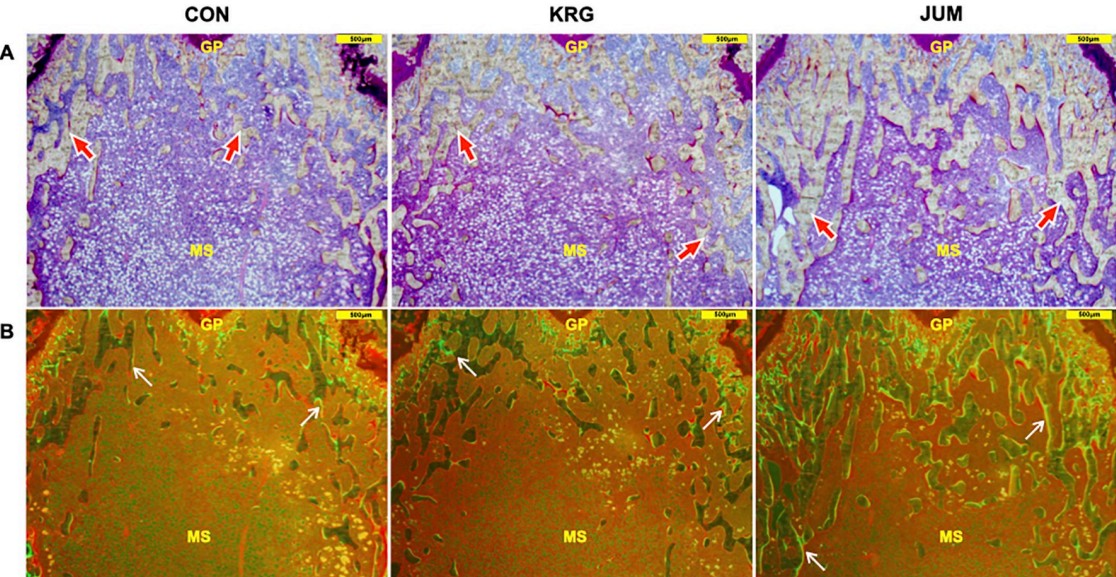

**Fig 5. Histological images of the distal femoral metaphyseal region from each group.** A) Representative sections of distal femurs under natural light. Red arrows show bone trabeculae. B) Representative sections of distal femurs under fluorescent light. The labeling surface with calcein is indicated by white arrows. Sections were stained by Villanueva staining. GP, growth plate; MS, marrow space; Scale bars (yellow) = 500 μm. KRG, group administered 200 mg/kg body weight of Korean red ginseng daily; JUM, group that jumped 10 times/day from a height of 40 cm; CON, control group without exercise or KRG administration.

**Table 2. Static and dynamic bone histomorphometric parameters of the distal femoral metaphysis.**

| | CON | KRG | JUM |
|---|---|---|---|
| BV/TV (%) | 12.1 ± 3.2 | 18.7 ± 3.9** | 22.4 ± 4.4*** |
| Tb.Th (μm) | 60.7 ± 9.4 | 70.2 ± 5.9 | 72.3 ± 7.2** |
| Tb.N (N/mm) | 2.0 ± 0.3 | 2.7 ± 0.5** | 3.1 ± 0.4*** |
| Tb.Sp (μm) | 456.2 ± 79.4 | 308.3 ± 58.1*** | 256.1 ± 46.6*** |
| Ob.S/BS (%) | 21.8 ± 7.4 | 23.3 ± 5.2 | 22.6 ± 2.7 |
| MS/BS (%) | 39.1 ± 4.5 | 36.2 ± 4.1 | 38.6 ± 1.5 |
| N.Ob/BS (N/mm) | 15.0 ± 5.1 | 18.8 ± 5.3 | 19.3 ± 5.7 |
| Oc.S/BS (%) | 13.4 ± 2.8 | 8.5 ± 3.6** | 9.4 ± 2.5* |
| ES/BS (%) | 32.8 ± 5.4 | 26.1 ± 7.6* | 22.1 ± 4.5** |
| N.Oc/BS (N/mm) | 4.6 ± 1.0 | 2.7 ± 1.1*** | 3.2 ± 0.7** |
| MAR (μm/day) | 2.8 ± 0.2 | 2.7 ± 0.2 | 2.8 ± 0.1 |
| BFR/BS ($\mu m^3/\mu m^2$/day) | 1.09 ± 0.14 | 0.97 ± 0.14 | 1.07 ± 0.06 |

Values are shown as means ± SD. BV/TV, trabecular bone volume fraction; Tb.Th, trabecular thickness; Tb.N, trabecular number; Tb.Sp, trabecular separation; Ob.S/BS, osteoblast surface per BS; MS/BS, mineralizing surface per BS; N.Ob/BS, number of osteoblasts per BS; Oc.S/BS, osteoclast surface per BS; ES/BS, eroded surface per BS; N.Oc/BS, number of osteoclasts per BS; MAR, mineral apposition rate; BFR/BS, bone formation rate per BS. Male Wistar rats that were 11 weeks old at the beginning of the study were distributed into 3 groups: a group administered 200 mg/kg body weight of Korean red ginseng daily (KRG); a group that jumped 10 times/day from a height of 40 cm (JUM); and control group without exercise or KRG administration (CON). Significant difference vs. CON group:

*$p < 0.05$,

**$p < 0.01$,

***$p < 0.001$.

trabecular thickness, and a 14% increase in trabecular number. In the bone histomorphometric analysis of the left femur distal metaphysis, the results of static histomorphometry are very similar to data from 3D morphometric parameters of current study.

In contrast to jump exercise, however, administration of KRG extract increased trabecular number (13%) when compared with the sedentary control group, but trabecular thickness was unaffected. These results imply that the trabecular bone gain induced by KRG administration is mainly due to increases in trabecular number as opposed to trabecular thickness. Several previous studies support this finding. For example, Kim et al. showed that *Panax ginseng* prevented age-related decreases in trabecular number without significant effects on trabecular thickness in a rat model [34]. Lee et al. also showed that oral administration of KRG extract to irradiated mice for 12 weeks prevented loss of trabecular bone mass of the proximal tibia, primarily by significant alterations in trabecular number [23]. Furthermore, we have previously demonstrated that both running and jump exercises exerted different effects on trabecular microarchitecture in the distal metaphysis in rats while both of them similarly enhanced trabecular bone mass [22, 26]. Taken together, these results suggest that KRG and jump exercises may produce trabecular bones with different structural and mechanical characteristics.

Bone mineral density is a major determinant of bone strength, but it does not capture all aspects that define the various mechanical behavior of trabecular bone. In addition to bone mass, microarchitecture also play an important role in determining the mechanical properties of trabecular bone [35]. In this study, several non-metric parameters characterizing bone architecture were also evaluated in addition to trabecular thickness, number and separation. Among these non-metric parameters Conn.D was significantly higher in the KRG group compared with CON, whereas TBPf and SMI showed significant changes in favor of increased bone strength in the JUM group. These results suggest that KRG and jump exercise may exert

different effects on the structural and mechanical characteristics of trabecular bone. Although the fracture load predicted by FEA was higher in both KRG and JUM groups compared with the CON group, the difference was significant only in the JUM group. Since the magnitude of increase in bone mass was smaller in the KRG group than in the JUM group, the statistical power of the present investigation may have been insufficient to detect differences in fracture load between KRG and CON.

Several studies have demonstrated that ginsenosides inhibit osteoclastogenesis and reduce bone resorption [15–18], although this is not uniformly the case [36, 37]. The histomorphometric data in the present study showed similar results. Osteoclastic parameters such as Oc.S/ BS, ES/BS, and N.Oc/BS were inhibited more by KRG, without changes in osteoblastic parameters. A previous study found that the restoration of trabecular bone architecture induced by treatment with an osteoclastic inhibitor (tiludronate) during remobilization after suspension-induced osteopenia in rats was predominantly attributable to increased trabecular number [38], and the present results are similar to this. From this perspective, the data obtained in the present study imply that the effect of KRG extract in increasing trabecular number was predominantly mediated by inhibition of bone resorption rather than stimulation of bone formation. On the other hand, the mechanical loading induced by physical exercise is well known to increase bone formation and reduce bone resorption, leading to the maintenance of a healthy skeleton [39]. In addition, jump exercise that principally promotes osteogenesis of osteoblasts have been reported to increase trabecular thickness of the distal femoral metaphysis in rats, but not trabecular number, resulting in an elevated trabecular bone volume [26]. The upshot of the present study was that jump exercise can significantly increase trabecular bone thickness by inhibiting bone resorption and increasing bone formation. Further synergistic effects are expected when KRG is used in combination with jump exercise, because KRG extract and jump exercises may change the trabecular bone architecture via different mechanisms of action.

Some limitations to the animal model used and the length of KRG administration applied during this study must be considered when interpreting the results. First, although there are similarities in bone metabolism between rats and humans, the inherent differences in bone structure, remodeling, and skeletal loading patterns should be considered before extrapolating our results to humans. Second, evaluation of the KRG effect on improvements in trabecular microarchitecture was performed at only a single dose selection (dose of 200mg/kg) and one time point (6 weeks after treatment). A longer administration duration and higher KRG dose could have lead to a greater increase in trabecular bone mass in KRG group. Further studies are warranted to confirm whether a dose higher than 200mg/kg and long-term use would provide better improvements to trabecular bone microarchitecture. However, maximizing this response was not the main purpose of the present study. Nonetheless, the dose and duration of KRG extract administration selected for the present study was sufficient to achieve increases in trabecular bone volume in the distal femoral metaphysis of growing male rats. Third, in this study, in order to clarify the relationship between trabecular bone structure and bone strength, FEA was performed using a model in which only the trabecular bone region was extracted. Therefore, the mechanical condition in the FEM model may be slightly different from the actual anatomical condition. Finally, this study did not include the combination of KRG and jump exercise. The present study showed that KRG has a weak but significant positive effect on bone mass and further suggest that the effect on trabecular microarchitecture is different from that by jump exercise. Thus the combination of KRG and jump exercise may synergistically lead to increase in trabecular bone mass and strength.

In conclusion, we demonstrated that the effects on trabecular bone mass differed between jump exercise and KRG, in that jump exercise increased trabecular bone volume by thickening

trabeculae, whereas administration of KRG extract added to trabecular bone volume by increasing trabecular number without accompanying increases in trabecular thickness. Whether combining KRG and jump exercise could increase trabecular bone mass and strength in an additive or even synergistic manner warrants investigation, as the mechanisms of bone response differed. Further studies are required to clarify this relationship. Taken together, our data may provide basic insights into the efficacy of KRG extract on trabecular bone architecture, supporting its development as a safe therapeutic agent and functional food with beneficial effects on bone.

## Supporting information

**S1 Dataset.**
(XLSX)

## Acknowledgments

This study was supported by Grant-in-Aid for Scientific Research (C) No. 15K01738 from the Japan Society for the Promotion of Science.

## Author Contributions

**Conceptualization:** Yong-In Ju, Teruki Sone.

**Data curation:** Yong-In Ju, Teruki Sone.

**Formal analysis:** Yong-In Ju, Hak-Jin Choi.

**Funding acquisition:** Yong-In Ju.

**Investigation:** Yong-In Ju.

**Methodology:** Yong-In Ju, Hak-Jin Choi.

**Project administration:** Yong-In Ju.

**Supervision:** Yong-In Ju, Teruki Sone.

**Validation:** Yong-In Ju.

**Visualization:** Yong-In Ju.

**Writing – original draft:** Yong-In Ju.

**Writing – review & editing:** Teruki Sone.

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
