## [Decision Letter · Decision Letter 0]

3 Dec 2021

PONE-D-21-20663Effects of Korean red ginseng on three-dimensional trabecular bone microarchitecture and strength in growing rats: comparison with changes due to jump exercisePLOS ONE

Dear Dr. Yong-In Ju,

Thank you for submitting your manuscript to PLOS ONE. After careful consideration, we feel that it has merit but does not fully meet PLOS ONE’s publication criteria as it currently stands. Therefore, we invite you to submit a revised version of the manuscript that addresses the points raised during the review process.

We look forward to receiving your revised manuscript.

Kind regards,

Ewa Tomaszewska, DVM Ph.D

Academic Editor

PLOS ONE

Journal Requirements:

2. In your Methods, please provide the product number for the KRG extract you obtained from Korea Ginseng Corporation, and include details of any quality assessments provided.

3. Please provide details about any efforts to minimise distress during the jumping task.

4. To comply with PLOS ONE submissions requirements, please provide the method of euthanasia in the Methods section of your manuscript.

5. As part of your revision, please complete and submit a copy of the Full ARRIVE 2.0 Guidelines checklist, a document that aims to improve experimental reporting and reproducibility of animal studies for purposes of post-publication data analysis and reproducibility: https://arriveguidelines.org/sites/arrive/files/Author%20Checklist%20-%20Full.pdf (PDF). Please include your completed checklist as a Supporting Information file. Note that if your paper is accepted for publication, this checklist will be published as part of your article.

6.Please state specifically whether the animal ethics committee specifically approved the study.

Reviewers' comments:

Reviewer's Responses to Questions

**Comments to the Author**

1. Is the manuscript technically sound, and do the data support the conclusions?

Reviewer #1: Partly

Reviewer #2: Partly

2. Has the statistical analysis been performed appropriately and rigorously? 

Reviewer #1: Yes

Reviewer #2: I Don't Know

3. Have the authors made all data underlying the findings in their manuscript fully available?

Reviewer #1: No

Reviewer #2: No

4. Is the manuscript presented in an intelligible fashion and written in standard English?

Reviewer #1: Yes

Reviewer #2: Yes

5. Review Comments to the Author

Reviewer #1: Reviewer's comments on paper “Effects of Korean red ginseng on three-dimensional trabecular bone microarchitecture and strength in growing rats: comparison with changes due to jump exercise” submitted to Plos One (PONE-D-21-20663).

Although some interesting results seems to be reported in this work, Authors didn't draft the manuscript well and I cannot recommend the publication of the article in its present form.

The (L24) abstract and introduction (L49-59) suggest that the aim of the study was to compare the effect of KRG with Jump-exercise in rat osteoporosis model. There is no explanation how 11 wk-old (male rats can serve as a osteoporosis model animal - no osteoporosis agent was introduced to animals in this study (ovariectomy, orchidectomy, hypophysectomy, parathyroidectomy, immobilization, or dietary manipulation). In rats, first osteoporotic changes in long bone metaphysis occurs in older animals, peak bone mass in trabeculae of femur metaphysis is reached at the age of ca. 6 months. On the other hand, the last sentences of the introduction indicate that the true aim of the study was to assess the influence of KRG as an effective strategy for improving bone strength during growth (L91) and compare this effect with well-known method of stimulation of bone formation in young adults – exercises. Therefore, I suggest the correct the manuscript by removing most of the osteoporosis-related fragments and make it cleaner that this study is focused on the young individuals during development.

There is only one dose examined, which selectin was based on a single reference data in which a different osteoporosis model (OVX) was under study. In this manner, the idea comparison of KRG (dietary supplement) with jump-exercised group is not sufficiently explained.

Limitations of the study should be extended as there are three main limitation of this study: the mentioned issue with the single-dose selection (and a single jump regime in lesser extent), the lack of the KRG+JUM group and the single time-point analysis.

Distal femoral histomorphometry was performed both using uCT and histological analysis. Were these results somehow consistent? It should be discussed.

Minor comments (selected):

L31 the information about KRG form is missing – extract? powder?

L53 the direct reference to WHO should be given (but I recommend remove all fragments related to primary osteoporosis)

L53 “bone destruction” – rephrase

L85 add information that this correspond to “rat model studies” as all references [25-27] are rat-related

L117 was it pure EtOH or phosphate-buffered EtOH?

L142 copper

L157 what was the mineral density of hydroxyapatite phantom

L173-174 I think that this information should be placed earlier, when animal management is described

L178 Correct to Villanueva osteochrome bone stain. Also was it ready-to-use kit ? If yes, please give the information of manufacturer.

L179-186 How all of these indices were calculated? what software was used ?

Statistics: As you present the results of statistical analysis as a single comparison of KGR group vs CON or vs JMP, why didn't you perform simple t-test or contrast analysis of KGR vs appropriate group instead of Tukey’s test where all three groups are examined ?

L203 What was the idea of presenting the weight of the calf muscle ? Was the increase of muscle weight expected (any references?)? Also, why tight muscle, which is more appropriate for any femur analysis than calf, was not examined? Finally, if some changes in muscle structure were expected, muscle histology also should be performed.

L301 hPTH? What is the connection of human parathyroid hormone with current study ?

Fig 2 - No A, B ... H on figure 2.

Fig 5 - more interesting than showing trabeculae would be showing calcenin labeling lines

which are barely visible in presented figures

Reviewer #2: Review of PONE-D-21-20663

Effects of Korean red ginseng on three-dimensional trabecular bone microarchitecture and strength in growing rats: comparison with changes due to jump exercise

General Comments

This study examined the effects of KRG on the trabecular bone of the distal femur in young growing male Wistar rats. Trabecular architecture and dynamic histomorphometry were employed as primary measures, complementing with computational models of the extracted trabecular architecture. Only the distal femur trabecular compartment was measured, with no axial skeleton examined and no diaphyseal cortex examined. As a comparator, the jump exercise model was examined in a matched cohort of rats, demonstrating a larger positive effect on this compartment. Both models suggested a shift to increased osteoblast numbers and decreased osteoclast numbers. The typical combined therapeutic (KRG + exercise) was not examined in this study, but has likely been pursued by this group (I encourage inclusion of such data in this paper if available).

Many details regarding the computational models are not included and need to be clarified in the manuscript. Some additional computational series should be run to answer the questions posed in the specific comments below.

Jump exercise had a measured effect of reducing femur length by 0.6-0.7 mm, which is a substantial amount. As an initial bound on this examination, t-tests performed without statistical penalty for JUM vs KRG and JUM vs CON demonstrate p values of 0.014 and 0.057, substantially more indicative of real difference than afforded by osteoblast numbers that are noted in presentation of results with p values > 0.5 due to variation. What occurs with jump exercise to stunt longitudinal growth? Is there damage to the physis?

The importance of ginseng processing is detailed, but the processing and verification of constituents and potency of the commercial product used in this study were not measured or reported. A test of residual KRG product should be performed prior to publication of this study.

The study would be improved with the inclusion of measurements for the axial skeleton (e.g. lumbar vertebrae) and cortical bone (e.g., femoral mid-diaphysis cortex) as is customary for bone morphometry measurements in studies of this nature. These data should be included for publication.

Specific Comments

Line 33: State the actual estimated jump height, and not the box height of 40 cm, as that seems misleading. See comment below regarding line 122.

Lines 75-77: These studies thus implicate direct action on osteoclasts. Did any of these in vivo studies perform histological measurements?

Lines 93-94: Cite references that provide data demonstrating this conclusion of ideal.

Line 109: Should “anesthetized” be “euthanized”?

Line 117: Why are no histological data reported in the abstract methods and results? They are very important.

Line 122: As described with a 40 cm tall box, how high must the rats jump from standing stretched position in order to grasp the box wall? It seems that the jump height would be 10-20 cm. Please clarify in the manuscript.

Lines 168-169: The method and algorithm implemented for assigning elastic modulus based on micro-CT derived tissue mineral density (using the Carter & Hayes relationship) must be provided. Was there any volumetric averaging of discrete values or was every voxel assigned a unique value? (unique assignment to each voxel does not seem justified)? Are your results different if one homogeneous value of elastic modulus is assigned to all regions of all models? Typically, trabecular structure dominates the variance in modulus assigned via such mineralization rules. Most importantly, what failure criterion was implemented for strength?

Lines 129-130: Measurements should have been performed to assess the true composition of ginsenosides, etc. within this commercial sample of KRG. Very frequently the true composition of such products is not as stated on the label.

Line 166: The boundary condition of uniform applied force rather than uniform displacement (Fig 1E) does not seem realistic, given the heterogeneous assignment of modulus values. That is, the lack of geometrical constraint due to extraction of trabecular bone, along with the spatial difference in elastic modulus, would seemingly allow each boundary location of applied force to deform differentially and unrealistically relative to the anatomical condition.

Line 166-168: What is the basis for selecting 2.8% as a threshold for defining failure. How do the results change if that value is perturbed upward and downward? Most importantly, what material failure criterion was applied in the definition of failure? This is a critical part of this method.

Lines 216-217: The stated differences in vBMD of 2% and 4% for jump exercise relative to control and KRG do not match the bar charts in Figure 2, which show large percentage differences similar to BV/TV (as they should). The data in figure 2 are almost certainly values of vBMD (i.e., apparent density) and report essentially the same comparative data as BV/TV. Presumably this statement regarding 2-4% pertains to the tissue-level density, which typically varies at most by a few percent. However, the manuscript makes no reference to tissue mineral density measurement throughout.

Lines 237-238: It does appear that osteoblast numbers were 25-30% higher in the experimental groups, but that measurement variation swamped the statistical comparison. It might be more insightful to the reader to provide the information in that manner.

Table 2: The histological data indicate that KRG lowered BFR/BS relative to control and jump exercise. Is this secondary to depletion of osteoclasts?

6. PLOS authors have the option to publish the peer review history of their article (what does this mean?). If published, this will include your full peer review and any attached files.

Reviewer #1: No

Reviewer #2: No

---

## [Author Response · Author response to Decision Letter 0]

24 Feb 2022

February 25, 2022

Dear. Ewa Tomaszewska

Academic Editor 

PLOS ONE

Manuscript ID PONE-D-21-20663

My colleagues and I wish to thank you and the reviewers for analysis of our manuscript entitled “Effects of Korean red ginseng on three-dimensional trabecular bone microarchitecture and strength in growing rats: comparison with changes due to jump exercise” (PONE-D-21-20663).

We have followed the suggestions of the editor and the reviewers and have made the following revisions:

Journal Requirements:

We have confirmed that our manuscript meets PLOS ONE’s style requirements.

2. In your Methods, please provide the product number for the KRG extract you obtained from Korea Ginseng Corporation, and include details of any quality assessments provided.

We have added accordingly (page 8, lines 154-157). 

3. Please provide details about any efforts to minimise distress during the jumping task.

We have added accordingly (page 8, lines 139-145).

4. To comply with PLOS ONE submissions requirements, please provide the method of euthanasia in the Methods section of your manuscript.

We have added accordingly (page 6, lines 106-111).

5. As part of your revision, please complete and submit a copy of the Full ARRIVE 2.0 Guidelines checklist, a document that aims to improve experimental reporting and reproducibility of animal studies for purposes of post-publication data analysis and reproducibility: https://arriveguidelines.org/sites/arrive/files/Author%20Checklist%20-%20Full.pdf (PDF). Please include your completed checklist as a Supporting Information file. Note that if your paper is accepted for publication, this checklist will be published as part of your article.

We have included the ARRIVE Guidelines Checklist accordingly.

6.Please state specifically whether the animal ethics committee specifically approved the study.

We have added accordingly (page 6, lines 103-106).

Responses to the Reviewers 

Reviewer #1

Comments to the Author

The (L24) abstract and introduction (L49-59) suggest that the aim of the study was to compare the effect of KRG with Jump-exercise in rat osteoporosis model. There is no explanation how 11 wk-old (male rats can serve as a osteoporosis model animal - no osteoporosis agent was introduced to animals in this study (ovariectomy, orchidectomy, hypophysectomy, parathyroidectomy, immobilization, or dietary manipulation). In rats, first osteoporotic changes in long bone metaphysis occurs in older animals, peak bone mass in trabeculae of femur metaphysis is reached at the age of ca. 6 months. On the other hand, the last sentences of the introduction indicate that the true aim of the study was to assess the influence of KRG as an effective strategy for improving bone strength during growth (L91) and compare this effect with well-known method of stimulation of bone formation in young adults – exercises. Therefore, I suggest the correct the manuscript by removing most of the osteoporosis-related fragments and make it cleaner that this study is focused on the young individuals during development.

In accordance with the reviewer’s comment, the manuscript has been revised (page 2, line 24 and page 4, lines 51-58).

There is only one dose examined, which selectin was based on a single reference data in which a different osteoporosis model (OVX) was under study. In this manner, the idea comparison of KRG (dietary supplement) with jump-exercised group is not sufficiently explained.

According to our knowledge, the effect of KRG extract on trabecular bone microarchitecture has not been verified in growing rat. Thus, we conducted explorative experiment at different dose of KRG extract (100mg/kg, 200mg/kg and 300mg/kg) using the smallest sample size for each condition before starting the present study. Bone mass in both dose of 200mg/kg and 300mg/kg showed a greater increase than in the dose of 100mg/kg, but no difference was observed between the dose of 200mg/kg and 300mg/kg. A similar effect has also been described by Avsar et al., (Cell Mol Biol. 2013;59(Suppl):OL1835-1841) who reported that BMD in 200mg/kg administered group showed a greater increase than 100ng/kg administered group. Thus, using data from explorative experiment in our laboratory and other studies, we determined the dose of KRG administration required for the experiment.

Comments on the idea comparison of KRG with jump exercise have been added to the manuscript (page 5, lines 88-92).

Limitations of the study should be extended as there are three main limitation of this study: the mentioned issue with the single-dose selection (and a single jump regime in lesser extent), the lack of the KRG+JUM group and the single time-point analysis.

In accordance with the reviewer’s suggestion, comments on limitations of the study have been added to the manuscript (pages 17-18, lines 344-350 and lines 357-361).

In the study by Umemura et al (J Bone Miner Res. 1997;12:1480-1485), they showed that although a slight tendency toward increase according to the number of jumps per day was observed, there were few differences in bone morphological and mechanical parameters among the 5-, 10-, 20-, and 40-jump groups. In this study, therefore, we did not consider the dose-effect in jump exercise.

Distal femoral histomorphometry was performed both using uCT and histological analysis. Were these results somehow consistent? It should be discussed.

We have added accordingly (page 15, lines 288-290).

Minor comments (selected):

L31 the information about KRG form is missing – extract? powder?

We have added accordingly (page 2, line 31). 

L53 the direct reference to WHO should be given (but I recommend remove all fragments related to primary osteoporosis)

We have modified accordingly (page 4, lines 51-58).

L53 “bone destruction” – rephrase

We have modified accordingly (page 4, lines 51-58).

L85 add information that this correspond to “rat model studies” as all references [25-27] are rat-related

The information regarding "rat model studies" has been added to the manuscript (page 5, lines 84-85).

L117 was it pure EtOH or phosphate-buffered EtOH?

It was pure EtOH. We have added accordingly (page 11, line 201). 

L142 copper

We have corrected a typing error "coper" to "copper" (page 9, line 166).

L157 what was the mineral density of hydroxyapatite phantom

The information regarding the mineral density of hydroxyapatite phantom has been added to the manuscript (page 10, line 180).

L173-174 I think that this information should be placed earlier, when animal management is described

The manuscript has been revised as suggested (page 7, lines 122-124).

L178 Correct to Villanueva osteochrome bone stain. Also was it ready-to-use kit ? If yes, please give the information of manufacturer.

The information of manufacturer has been added to the manuscript (page 11, lines 204-206). 

L179-186 How all of these indices were calculated? what software was used ?

The information about histomorphometric image analysis system has been added to the manuscript (page 11, lines 208-209).

Statistics: As you present the results of statistical analysis as a single comparison of KGR group vs CON or vs JMP, why didn't you perform simple t-test or contrast analysis of KGR vs appropriate group instead of Tukey’s test where all three groups are examined ?

In this study, instead of simple pairwise comparison with CON, we adopted a more robust method of ANOVA and post-hoc analysis among three equivalent groups.

L203 What was the idea of presenting the weight of the calf muscle ? Was the increase of muscle weight expected (any references?)? Also, why tight muscle, which is more appropriate for any femur analysis than calf, was not examined? Finally, if some changes in muscle structure were expected, muscle histology also should be performed.

Change in muscle mass may play an important role in the regulation of bone mass. However, there was no significant difference in muscle weight between the groups. This phenomenon is not unusual in studies using rats. The comparability of muscle weight between the study groups strengthens the conclusion that the bone changes observed in our study are derived primarily from the exercise per se.

The anatomy of the thigh muscles is markedly different in rat and human, with the biceps femoris being a far larger muscle in rat, with an extensive attachment along the tibia. Therefore, it is very unlikely that these large muscles will increase muscle weight through 10 time jump exercise protocol. Furthermore, since the removal procedure of large muscles such as biceps femoris, vastus medialis, and rectus femoris of rat is more complicated than the simple extraction of calf muscles, it could be an error factor in the comparison of muscle weight.

L301 hPTH? What is the connection of human parathyroid hormone with current study?

Since the direct relationship with the results of this study is small, we have deleted the word of hPTH (page 17, line 332).

Fig 2 - No A, B ... H on figure 2.

We have added accordingly (Figure 2). 

Fig 5 - more interesting than showing trabeculae would be showing calcenin labeling lines

which are barely visible in presented figures

We much appreciate the reviewer’s suggestion. We have modified accordingly (page 26, lines 524-526; Figure 5).

Reviewer: 2

Comments to the Author

General Comments

This study examined the effects of KRG on the trabecular bone of the distal femur in young growing male Wistar rats. Trabecular architecture and dynamic histomorphometry were employed as primary measures, complementing with computational models of the extracted trabecular architecture. Only the distal femur trabecular compartment was measured, with no axial skeleton examined and no diaphyseal cortex examined. As a comparator, the jump exercise model was examined in a matched cohort of rats, demonstrating a larger positive effect on this compartment. Both models suggested a shift to increased osteoblast numbers and decreased osteoclast numbers. The typical combined therapeutic (KRG + exercise) was not examined in this study, but has likely been pursued by this group (I encourage inclusion of such data in this paper if available).

In this study, we elucidated the differential effects of jump and KRG on trabecular bone architecture in rats. Thus the combination of KRG and jump exercise may synergistically lead to increase in trabecular bone mass and strength. We are currently conducting additional experiments to find out about that.

Comments on the lack of the combination of KRG and jump exercise have been added to the manuscript (pages 17-18, lines 353-357).

Many details regarding the computational models are not included and need to be clarified in the manuscript. Some additional computational series should be run to answer the questions posed in the specific comments below.

Jump exercise had a measured effect of reducing femur length by 0.6-0.7 mm, which is a substantial amount. As an initial bound on this examination, t-tests performed without statistical penalty for JUM vs KRG and JUM vs CON demonstrate p values of 0.014 and 0.057, substantially more indicative of real difference than afforded by osteoblast numbers that are noted in presentation of results with p values > 0.5 due to variation. What occurs with jump exercise to stunt longitudinal growth? Is there damage to the physis?

According to our previous studies (Bone. 2003;33:485-493, J Appl Physiol. 2008;104:1594-1600, 2012;112:766-772, 2015;119:990-997, PLoS One. 2014;9:e107953, Phys Act Nutr. 2020;24:1-8), the decrease in femoral length was not always by exercise stress. Various factors can be considered as the cause such as intake of food and water, body weight, breeding conditions, and measurement variation. Even if some impaired longitudinal bone growth occurs by these factors, the influence of these factors is even among the experimental groups. Although we cannot completely exclude the possibility of the influence of jump exercise on the longitudinal bone growth, we believe the effects would be negligible, if any.

The importance of ginseng processing is detailed, but the processing and verification of constituents and potency of the commercial product used in this study were not measured or reported. A test of residual KRG product should be performed prior to publication of this study.

The composition of ginsenosides and a broad range of efficacy of the KRG extract used in this study have been approved by the Korean Food and Drug Administration (KFDA), and we contacted the Korea Ginseng Corporation directly and confirmed the true composition of ginsenosides.

We have added accordingly (page 8, lines 154-157).

The study would be improved with the inclusion of measurements for the axial skeleton (e.g. lumbar vertebrae) and cortical bone (e.g., femoral mid-diaphysis cortex) as is customary for bone morphometry measurements in studies of this nature. These data should be included for publication.

It has been reported that bone mass and strength in the lumbar vertebrae of rat are not significantly affected by exercise (Iwamoto et al., Bone. 1999;24:163-169, J Bone Miner Metab. 2004;22:26-31, Notomi et al., J Appl Physio. 2002;93:1152-1158). Also, in the previous report, exercise showed a positive effect on bone mass in long bones at weight-bearing sites such as tibia and femur compared with vertebral bone. Furthermore, the adaptive response of bone to exercise differs between regions of the same bone because the stress distribution by exercise differ among skeletal sites. In our laboratory’s previous studies in rats (data not presented), both jump and running exercises had less impact on cortical bone structural changes. For these reasons, axial skeleton and cortical bone were not included in the present study. The analysis of these sites would be included in future studies.

Specific Comments

Line 33: State the actual estimated jump height, and not the box height of 40 cm, as that seems misleading. See comment below regarding line 122.

The actual estimated jump height has been included in the manuscript (pages 7-8, lines 133-136).

Lines 75-77: These studies thus implicate direct action on osteoclasts. Did any of these in vivo studies perform histological measurements?

These previous studies have not been histologically examined and the relationship between KRG and bone histomorphometric parameters is unknown. We have modified the manuscript (page 5, lines 74-76). 

Lines 93-94: Cite references that provide data demonstrating this conclusion of ideal.

We have added accordingly (page 6, line 99).

Line 109: Should “anesthetized” be “euthanized”?

We have corrected accordingly (page 6, lines 106-111).

Line 117: Why are no histological data reported in the abstract methods and results? They are very important.

We have added accordingly (page 2, lines 35-36 and page 3, lines 42-43).

Line 122: As described with a 40 cm tall box, how high must the rats jump from standing stretched position in order to grasp the box wall? It seems that the jump height would be 10-20 cm. Please clarify in the manuscript.

The actual estimated jump height has been included in the manuscript (pages 7-8, lines 133-136).

Lines 168-169: The method and algorithm implemented for assigning elastic modulus based on micro-CT derived tissue mineral density (using the Carter & Hayes relationship) must be provided. Was there any volumetric averaging of discrete values or was every voxel assigned a unique value? (unique assignment to each voxel does not seem justified)? Are your results different if one homogeneous value of elastic modulus is assigned to all regions of all models? Typically, trabecular structure dominates the variance in modulus assigned via such mineralization rules. Most importantly, what failure criterion was implemented for strength?

We have added accordingly (page 10, lines 191-198).

Lines 129-130: Measurements should have been performed to assess the true composition of ginsenosides, etc. within this commercial sample of KRG. Very frequently the true composition of such products is not as stated on the label.

The composition of ginsenosides and a broad range of efficacy of the KRG extract used in this study have been approved by the Korean Food and Drug Administration (KFDA), and we contacted the Korea Ginseng Corporation directly and confirmed the true composition of ginsenosides.

We have added accordingly (page 8, lines 154-157).

Line 166: The boundary condition of uniform applied force rather than uniform displacement (Fig 1E) does not seem realistic, given the heterogeneous assignment of modulus values. That is, the lack of geometrical constraint due to extraction of trabecular bone, along with the spatial difference in elastic modulus, would seemingly allow each boundary location of applied force to deform differentially and unrealistically relative to the anatomical condition.

In accordance with the reviewer’s suggestion, comments on limitations of the study have been added to the manuscript (pages 17-18, lines 353-357).

Line 166-168: What is the basis for selecting 2.8% as a threshold for defining failure. How do the results change if that value is perturbed upward and downward? Most importantly, what material failure criterion was applied in the definition of failure? This is a critical part of this method.

In this study, we chose 2.8% as the threshold for defining destruction based on literature that failure load was defined as the load when 2% of the elements exceeded a strain of 0·007 using HR-pQCT images (Samelson et al., Lancet Diabetes Endocrinol. 2019;7:34-43). If it is determined as a fracture when one element is destroyed, an element that is destroyed by the model error may come out. To avoid this error, we had a few percent margin to meet the occurrence of a sure local fractures.

We have modified the manuscript accordingly (page 10, lines 196-198).

Lines 216-217: The stated differences in vBMD of 2% and 4% for jump exercise relative to control and KRG do not match the bar charts in Figure 2, which show large percentage differences similar to BV/TV (as they should). The data in figure 2 are almost certainly values of vBMD (i.e., apparent density) and report essentially the same comparative data as BV/TV. Presumably this statement regarding 2-4% pertains to the tissue-level density, which typically varies at most by a few percent. However, the manuscript makes no reference to tissue mineral density measurement throughout.

We much appreciate the reviewer’s careful review. We made a typo and corrected "2% and 4%" as "118% and 55%" (page 13, line 246). 

Lines 237-238: It does appear that osteoblast numbers were 25-30% higher in the experimental groups, but that measurement variation swamped the statistical comparison. It might be more insightful to the reader to provide the information in that manner.

We agree with your comment. Using data from previous studies in our laboratory and other studies, we determined the minimum number of animals required for the experiment. However, the statistical power seems to be insufficient for some evaluation items, such as the results of static and dynamic histomorphometric. Perhaps a larger sample size may confirm these findings.

Table 2: The histological data indicate that KRG lowered BFR/BS relative to control and jump exercise. Is this secondary to depletion of osteoclasts?

We agree with your comment. Since both bone resorption (Oc.S/BS(%), ES/BS(%), and N.Oc/BS) and BFR/BS were lower, it is considered that the bone metabolism has shifted to low turnover rate. Namely, we speculate the low BFR/BS is secondary to the decrease of osteoclastic bone resorption by KRG.

We believe the thoughtful comments of the reviewers have substantially improved the manuscript. We hope that the manuscript will now be acceptable for publication in PLOS ONE.

We thank you very much for your kind consideration.

Cordially yours,

Yong-In Ju, Ph.D.

Kawasaki University of Medical Welfare

288 Matsushima, Kurashiki

Okayama 701-0193, Japan

Tel: +81-86-462-1111, Fax: +81-86-464-1109

E-mail: jyi.k051@mw.kawasaki-m.ac.jp

---

## [Decision Letter · Decision Letter 1]

11 Apr 2022

Effects of Korean red ginseng on three-dimensional trabecular bone microarchitecture and strength in growing rats: comparison with changes due to jump exercise

PONE-D-21-20663R1

Dear Dr. Yong-In Ju,

We’re pleased to inform you that your manuscript has been judged scientifically suitable for publication and will be formally accepted for publication once it meets all outstanding technical requirements.

Kind regards,

Ewa Tomaszewska, DVM Ph.D

Academic Editor

PLOS ONE

Additional Editor Comments (optional):

Reviewers' comments:

Reviewer's Responses to Questions

**Comments to the Author**

1. If the authors have adequately addressed your comments raised in a previous round of review and you feel that this manuscript is now acceptable for publication, you may indicate that here to bypass the “Comments to the Author” section, enter your conflict of interest statement in the “Confidential to Editor” section, and submit your "Accept" recommendation.

Reviewer #1: All comments have been addressed

2. Is the manuscript technically sound, and do the data support the conclusions?

Reviewer #1: Yes

3. Has the statistical analysis been performed appropriately and rigorously? 

Reviewer #1: Yes

4. Have the authors made all data underlying the findings in their manuscript fully available?

Reviewer #1: (No Response)

5. Is the manuscript presented in an intelligible fashion and written in standard English?

Reviewer #1: Yes

6. Review Comments to the Author

Reviewer #1: Dear authors,

The manuscript has been substantially improved, so my recommendation is to accept the manuscript in its present form.

7. PLOS authors have the option to publish the peer review history of their article (what does this mean?). If published, this will include your full peer review and any attached files.

Reviewer #1: No

---

## [Editor Report · Acceptance letter]

25 Apr 2022

PONE-D-21-20663R1 

Effects of Korean red ginseng on three-dimensional trabecular bone microarchitecture and strength in growing rats: comparison with changes due to jump exercise 

Dear Dr. Ju:

I'm pleased to inform you that your manuscript has been deemed suitable for publication in PLOS ONE. Congratulations! Your manuscript is now with our production department. 

Kind regards, 

on behalf of

Professor Ewa Tomaszewska 

Academic Editor

PLOS ONE